# A Dual Robust Strategy for Removing Outliers in Multi-Beam Sounding to Improve Seabed Terrain Quality Estimation

**DOI:** 10.3390/s24051476

**Published:** 2024-02-24

**Authors:** Ping Zhou, Jifa Chen, Shengping Wang

**Affiliations:** 1Research Center of Hydraulic Safety Engineering Technology, Jiangxi Academy of Water Science and Engineering, Nanchang 330029, China; pingzhou@cug.edu.cn; 2School of Hydraulic & Ecological Engineering, Nanchang Institute of Technology, Nanchang 330099, China; 3Key Laboratory of Poyang Lake Wetland and Watershed Research Ministry of Education, Jiangxi Normal University, Nanchang 330022, China; 4Key Laboratory of Submarine Geosciences, Second Institute of Oceanography, Hangzhou 310012, China; shpwang@ecit.edu.cn

**Keywords:** multibeam sounding, seabed terrain, robust estimation, polyhedral function, kriging algorithm

## Abstract

During the process of seabed terrain exploration using a multi-beam echo system, it is inevitable to obtain a sounding set containing anomalous points. Conventional methods for eliminating outliers are unable to reduce the disruption caused by outliers over the whole dataset. Furthermore, incomplete consideration is given to the terrain complexity, error magnitude, and outlier distribution. In order to achieve both a high-precision terrain quality estimate and quick detection of depth anomalies, this study suggests a dual robust technique. Firstly, a robust polyhedral function is utilized to solve anomaly detection for large errors. Secondly, the robust kriging algorithm is used for refined outlier removal. Ultimately, the process of dual detection and anomaly removal is achieved. The experimental results demonstrate that DRS technology has the most favorable mean square error and error fluctuation range in the test set, with values of 0.8321 and [−2.0582, 1.9209], respectively, when compared to RPF, WT, GF, and WLS-SVM schemes. Furthermore, DRS is able to adjust to various terrain complexities, discrete distribution features, and cluster outlier detection, as shown by objective indicators and visual outcome maps, guaranteeing a high-quality seabed terrain estimate.

## 1. Introduction

Seabed terrain data presents significant geomorphic features on the ocean’s surface, offering crucial data support for offshore operations, resource exploitation, benthic organism research, and foundation engineering building [1,2]. As a result, a lot of underwater measurements have been done, particularly since multi-beam echo sounders (MBESs) have become popular and provide an important means for ocean depth measurement. With its wide detection range and high data density, MBESs are able to collect high-resolution terrain data and complete coverage, providing an accurate description of the seafloor topography and geomorphometry [3,4]. However, affected by the complex marine hydrological environment, system interference, and ocean reverberation, the collected sounding data inevitably exhibits sharp and prominent outliers [5]. These abnormal values are evident in the seabed terrain map and easily lead to incorrect terrain information, which is very unfavorable for subsequent seabed exploration and engineering construction [6]. Consequently, the estimation of sounding quality greatly depends on effective and trustworthy outlier elimination techniques. It is vital to reduce these outliers’ interference to some extent while assessing underwater terrain and surface items.

The detection and removal of anomalous points in depth measurement is the basis for ensuring the quality estimation of water depth. Quality control is usually carried out from the levels of manual editing, waveform filtering, surface fitting, and robust estimation, as described in Table 1. Interactive manual editing of sounding data is commonly adopted, which is also the most challenging and time-consuming task [7]. Based on the statistical distribution rules of bathymetric data, Ladner et al. fitted and optimized estimates point by point within the grid to achieve automatic cleaning of abnormal sounding data [8]. However, the different distribution characteristics of grid intervals and interval points interfere with the accuracy of fitting estimation to a certain extent.

Waveform filtering detects spike signals with waveform changes in units of 1 Ping in depth measurement, which is utilized to separate anomalous sounding spots. In the early days, Ware et al. used the weighted moving average (WMA) method to detect anomalous values in depth measurement based on the assumption of stable fluctuations in the signal [9]. Bore et al. combined Gaussian filtering (GF) to eliminate local waveform anomalies in depth measurement [10]. In addition, Santos et al. employed wavelet transform (WT) to improve the quality of nearshore water depth estimation [11]. A quality factor forecasting error (QFQE) was presented by Zhou et al. to identify outliers and forecast depth in multi-beam sounding data [12]. QFQE approach found an outlier collection of sound points by fitting and estimating each sounding point using sliding windows and Kalman filtering. However, the accuracy of water depth estimate is connected to the parameters of the state transition matrix and process noise weight matrix in Kalman filtering. The experiment found that the efficiency of single Ping depth anomaly detection needs to be improved, and it is also necessary to combine the error judgment criteria in data statistics to achieve the determination of anomaly points.

The estimation method based on trend surface fitting came into being. The establishment of a trend surface assumes that the seabed topography changes continuously and gently, which can be approximated infinitely by using certain model functions. Niedzielski et al. utilized a quadratic polynomial to fit the bathymetric change trend [13]. The fitting coefficients were found with least squares. Based on the mean square error and residual error calculated by the model, it was judged whether the estimated bathymetry value is abnormal. The experimental results show that the accuracy of anomaly detection by polynomial fitting is low, and the estimated values vary greatly when the local shape fluctuates. Zhao et al. proposed a method combining uncertainty and bathymetry estimator (CUBE) to process raw MBES datasets with depths ranging from 10 to 11,000 m [14]. However, parameter optimization in combined water depth estimation is limited by the survey area, optimal grid resolution, and batch processing. In addition, Huang et al. combined the weighted least squares method and support vector machine (WLS-SVM) to construct the seafloor topographic surface to further detect water depth outliers [15]. LR B-splines allow iterative local optimization to construct approximate bathymetric surface models, thereby achieving full-span bathymetric refinement classification [16]. Experimental results indicated that the application of the B-splines model could effectively detect sounding outliers when the bathymetry conforms to the *t* distribution, and other distribution types need to be further verified.

Since the polyhedral function model can more accurately approximate complex terrain, it has been widely used in the construction of digital elevation model (DEM) [17]. Bao et al. adopted polyhedral function to construct a surface layer for analyzing the evolution of sedimentary structures and characterization of excess water pressure on the seabed surface with different particle sizes [18]. Welsch et al. built rock structures of submarine volcanoes with polyhedral function to reveal the characteristics and interactions of various rocks during volcanic eruptions [19]. Therefore, it is necessary to take the polyhedral function as the basis of bathymetric trend surface to reflect the variations in seabed relief.

Furthermore, robust estimation is applied by other notable algorithms to identify abnormal sounding points. Robust estimation serves to prevent outliers from interfering with the model’s overall training process. The basis for robust estimation is that the weight of normal bathymetry is 1, and the weight of outliers is close to 0. Debese et al. proposed a hierarchical adaptive robust method to construct the seafloor surface and isolate detection outliers [20]. Robust models such as Huber function [21], L1 norm [22], IGGIII estimator [23] and Tukey test [24] are utilized to analyze the outlier detection effect. The experimental results indicated that the IGGIII estimator and Tukey estimator had better performance indicators [25]. Therefore, IGGIII estimator will become the first choice for the weight function in robust estimation in this article. The weight discrimination of large bathymetry outliers is achieved through the least squares recursion method of weight selection iteration.

In addition, in order to identify smaller outlier locations and guarantee the continuity of seabed sounding, a variety of DEM interpolation construction techniques, including inverse distance weighting [26], nearest neighbor interpolation [27], and Kriging interpolation [28], have also been used to detect outliers and estimate the quality of depth measurements. Among them, kriging algorithm is used to take into account the spatial relationship between each sounding point and can obtain the optimal linear unbiased estimate of the variable. Kriging has higher valuation calculation accuracy and achieves better results [29]. However, if the abnormal values of bathymetry are directly treated as normal values for the calculation of the variation function, the variation function of kriging algorithm will deviate from the correct shape due to the influence of the abnormal values, resulting in a discrepancy between the drawn seabed terrain and the actual seabed. Therefore, it is necessary to improve the relevant modules of kriging algorithm.

**Table 1 sensors-24-01476-t001:** Statistical Methods for Eliminating Abnormalities in Sounding to Improve the Quality Estimation of Submarine Topography.

Aspects	Techniques	Characterization
Manual editing	Main software: Qimera Qinsy 9; Hypack 2024; Teledyne PDS2000; CARIS HIPS and SIPS version 11.1; MB System version 5.8.0.; etc. [30] ^1^	Time-consuming and rough elimination.
Waveform filtering	WMA [9]; GF [10]; WT [11]; Kalman filtering [12]; etc.	In units of Ping signal; Relatively low efficiency; Outliers of significant features.
Surface fitting	Polynomial fitting [13]; CUBE [14]; WLS-SVM [15]; B-splines [16]; Polyhedral function [17,18,19]; etc.	Simple structure; Outliers of significant features; Susceptible to outlier contamination.
Robust estimation	Huber function [21]; L1 norm [22]; IGGIII estimator [23,25]; Tukey test [24]; etc.	Pollution-resistant; High accuracy; Outliers of significant features.
Interpolation construction	inverse distance weighting [26]; nearest neighbor interpolation [27]; Kriging algorithm [28,29]; etc.	Simple structure; Requires high precision processing up front.

^1^ The provided information is the latest version of the software.

In order to get high precision estimation of seabed topography quality, a dual robust strategy (DRS) is developed in this study to address the issue of fine detection of sounding outliers. DRS technique includes the dual detection process of robust polyhedral function (RPF) and robust kriging algorithm (RKA). The polyhedral function in RPF model was employed to construct the bathymetric terrain trend surface. Combining the robust estimation model with iterative weight selection, the weight of outliers was continuously reduced to smoothly resist the interference of outliers on the overall terrain undulation. Furthermore, RKA approach was implemented to further solve the issue of local protruding anomaly characteristic by removing minor errors anomalous spots. Ultimately, the entire DRS technique is able to handle different types and sizes of sounding outliers to improve the quality assessment of underwater topography. The frame design for detecting and removing outliers in depth measurement is shown in Figure 1.

In this work, we have made the following progress: (1) A polyhedral function that is more in line with terrain fluctuations is applied to fit the trend term of sounding dataset; (2) Robust estimation is adopted to weaken the interference of abnormal sounding values on the overall fitting effect, so as to eliminate larger abnormal points; (3) Robust kriging algorithm is employed to further detect small outliers in the sounding.

The remainding of this paper is organized as follows. Section 2 introduces the proposed detection strategy for sounding outliers in detail. The experimental results are illustrated in Section 3. Different terrain categories are adopted to further verify the performance of the proposed technique, which is shown in the Section 4. Finally, the conclusions are presented in the Section 5.

## 2. Materials and Methods

### 2.1. Overall Framework

The seabed topography’s undulations are altered by aberrant bathymetry, which seriously obstructs seafloor exploration and scientific research and causes errors in underwater topography mapping [31,32]. For complex and undulating seafloor terrain areas, DRS technique is proposed to detect and eliminate bathymetric outliers, allowing for high-precision evaluation of terrain quality. The modules of this technique mainly include: (1) Trend surface construction with polyhedral function; (2) Weakening the weight of abnormal sounding values with robust iteration; (3) Eliminating small error outliers with RKA approach. The flow chart is shown in Figure 1.

### 2.2. Polyhedral Function and RPF Theory

#### 2.2.1. Polyhedral Function

(1)Polynomial Surface

It is assumed that the seabed terrain conforms to the characteristics of second-order variation, and the sounding data is a set of discrete observations conforming to a certain distribution. Based on a set of bathymetric values *ZZ* and plane coordinates (*x*, *y*) data, polynomial function $ZZ=f\left(x,y\right)$ uses surface fitting to approximate the spatial distribution of the terrain feature. The power degree of the polynomial is selected according to the actual degree of terrain relief, usually quadratic. It can be defined:(1)$$ZZ=f\left(x,y\right)=\theta_{0}+\theta_{1}x+\theta_{2}y+\theta_{3}x^{2}+\theta_{4}xy+\theta_{5}y^{2}$$
where *f*(*x*, *y*) is a general form of the function. *θ_i_* is fitting coefficient.

A series of sounding data are substituted into the general form of the function *f*(*x*, *y*) and expressed in matrix as:(2)$$A={\left[\begin{array}{llllll} 1 & x_{1} & y_{1} & x_{1}^{2} & x_{1}y_{1} & y_{1}^{2}\\ \cdots & \cdots & \cdots & \cdots & \cdots & \vdots \\ 1 & x_{m} & y_{m} & x_{m}^{2} & x_{m}y_{m} & y_{m}^{2} \end{array}\right]}_{m\times 6},\;\theta ={\left[\begin{array}{l} \theta_{0}\\ \vdots \\ \theta_{5} \end{array}\right]}_{m\times 1},\;ZZ={\left[\begin{array}{l} zz_{1}\\ \vdots \\ zz_{m} \end{array}\right]}_{m\times 1}$$
where *m* is the number of depth measurement datasets. *zz_i_* is the depth measurement value corresponding to coordinates (*x_i_*, *y_i_*). Matrix A are the polynomial function of coordinates (*x_i_*, *y_i_*). The elements of vector θ are the coefficient of the polynomial function, and the value can be obtained by $\theta ={\left(A^{T}A\right)}^{-1}A^{T}ZZ$.

The least square method is utilized to solve the parameters, and the outliers are calibrated according to the criterion of triple mean square error. The specific criterion is as follows:(3)$$\begin{array}{l} \left\{\begin{array}{l} \left|zz_{i}-f\left(x_{i},y_{i}\right)\right|\leq 3\sigma ,\;nomal\\ \left|zz_{i}-f\left(x_{i},y_{i}\right)\right|>3\sigma ,\;outlier \end{array}\right.\\ \sigma =\sqrt{\left(vv\right)/\left(n-1\right)} \end{array}$$
where σ is mean square error of depth measurement dataset. *v* is the residual of depth measurement dataset.

Experiments have found that substituting aberrant sounding points into polynomial fitting functions will significantly contaminate the overall model and result in mistakes in the parameters that are derived from the solution [15]. Furthermore, polynomial functions are not capable to accurately represent the real terrain indefinitely [12]. This model is typically appropriate for identifying locations with notable anomalies and slight variations in the topography.

(2)Polyhedral function model

The seafloor topography is uneven due to its undulations. When the undulations are severe, it is difficult for the polynomial function to accurately reflect the changing process of the seabed terrain. The trend surface constructed through polynomial function has resulted in a high collapse rate when using the obtained fitting residual value as the initial value for robust iteration [33]. However, the polyhedral function model could establish functional relationships between each sampling point and all known nodes, which approximated the real terrain through contribution values [17]. Taking polyhedral function as the basis function to construct the regional seabed terrain, the model is expressed as:(4)$$L\left(x,y\right)=\sum_{i=1}^{n}c_{i}R\left(x,y,x_{i},y_{i}\right)$$
where (*x*, *y*) is the coordinate of sampling point. *L*(*x*, *y*) is the sounding value of sampling point. *n* is the number of known points. (*x_i_*, *y_i_*) is the coordinate of known point. *R(x*, *y*, *x_i_*, *y_i_)* represents the functional relationship between a sampling point (*x*, *y*) and a known node (*x_i_*, *y_i_*). The undetermined coefficient of *c_i_* is the contribution of point *i* to the sampling point.

The kernel function *R* is usually symmetrical and expressed as:(5)$$R\left(x,y,x_{i},y_{i}\right)={\left[{\left(x-x_{i}\right)}^{2}+{\left(y-y_{i}\right)}^{2}+\delta \right]}^{\beta }$$
where *δ* is a smoothing factor. *β* is a power index and usually chosen as −0.5, 0.5, 1.5.

Based on the terrain and experimental conditions of the study area, the two elements of the polyhedral function model are: δ = 10,000 and *β* = 0.5. *n* is half of the experimental observation data. That is, half of the data points of the multi-beam bathymetric data are selected as known nodes. Formula (4) is written in matrix form as:(6)$$\left[\begin{array}{c} L_{1}\\ L_{2}\\ \vdots \\ L_{m} \end{array}\right]=\left[\begin{array}{cccc} R_{11} & R_{12} & \cdots & R_{1n}\\ R_{21} & R_{22} & \cdots & R_{2n}\\ \vdots & \vdots & \ddots & \vdots \\ R_{m1} & R_{m2} & \cdots & R_{mn} \end{array}\right]\cdot \left[\begin{array}{c} c_{1}\\ c_{2}\\ \vdots \\ c_{n} \end{array}\right]=L=RC$$
where L is the matrix of sounding dataset. *m* is the number of depth measurement datasets. *R* is a kernel function.

*C* is a generalized inverse solution. The parameter coefficients of the vector *C* can be derived by the least squares approach.
(7)$$V=RC\Rightarrow C={\left(R^{T}PR\right)}^{-1}R^{T}PL$$
where *P* is the weight of the observation value, and it is an equal weight observation during the initial iteration. The weight *P* of observation points are iteratively and constantly adjusted when robust estimation is implemented in the following phase.

The residual error Δ*_V_* = [Δ*_V_*_1_, …, Δ*_V__m_*] can be obtained by substituting the sounding values of waiting points into Equation (7). Δ*_V_* is represented as:(8)Δv=Z−RC
where *Z* is the matrix of the depth measurement dataset that needs to be judged.

A simple polyhedral function model is just that. Prior to implementing a robust estimation model, the sounding residual Δ*_V__i_* and the criterion of triple mean square error were applied to identify outliers.

Naturally, it has been discovered that the random selection of known nodes eventually adds data from outlier sites to the model through the use of a polyhedral function model. The quality of the model’s depth estimation and judgment is impacted by information pollution from outlier locations in the basic version of the polyhedral function.

#### 2.2.2. RPF Theory

We can find that when selecting bathymetry data as known nodes, a small number of outliers are inevitably taken as known observation signals. Robust estimation can effectively weaken the interference of these outliers on the overall data without affecting subsequent data processing. The idea of robust estimation is to continuously optimize the weight of observation values through the process of robust iteration [34]. The weight of outliers is reduced to zero to detect and eliminate outliers. The iterative solution of step *k* + 1 is:(9)$${\overset{\wedge }{C}}^{\left(k+1\right)}={\left(R^{T}{\bar{P}}^{\left(k\right)}R\right)}^{-1}R^{T}{\bar{P}}^{\left(k\right)}L$$
where $\bar{P}=\left[\begin{array}{lll} {\bar{p}}_{i} & \cdots & {\bar{p}}_{m} \end{array}\right]$ is the equivalent weight function, consisting of the diagonal matrix of the sounding observations. $\bar{P}$ is the weighted form of *P* in Equation (7). *m* is the number of depth measurement datasets.

The IGGIII estimator is a related equivalent weight function proposed by Yang [35]. Different weight functions and tolerance criteria are applied for different measurement data, with the objective being to completely account for the real circumstances of the measurement data. Three categories are used to group the measurement data: normal, available, and hazardous. Sounding outlier is classified as harmful data, which can interfere with the estimation quality of the model solution [23]. The normal data information should be fully utilized during the state estimation procedure, maintaining the original weights. Additionally, the weight of the available data keeps decreasing, lessening the interference influence. The process of identifying harmful data is carried out iteratively, using weights that are directly close to zero, so as not to impede the solution of model parameters. Its function can be stated as follows:(10)$${\bar{p}}_{i}=\left\{\begin{array}{cc} p_{i} & \left|{v_{i}}^{\prime }\right|<T_{1}\\ p_{i}\frac{T_{1}}{\left|{v_{i}}^{\prime }\right|}\left[\frac{T_{2}-\left|{v_{i}}^{\prime }\right|}{T_{2}-T_{1}}\right] & T_{1}\leq \left|{v_{i}}^{\prime }\right|<T_{2}\\ 0 & \left|{v_{i}}^{\prime }\right|>T_{2} \end{array}\right.$$
where ${\bar{p}}_{i}$ represents the weight during the iterative update of point *i*, *p_i_* is the weight of the previous generation, and the initial value is 1. $v_{i}^{\prime }$ is standardized residual error that represents the relationship between residual error and mean square error. *T*_1_ and *T*_2_ constants are standardized residual thresholds. The experimental results in literature [23] indicate that the values of *T*_1_ and *T*_2_ range from 1 to 2.5 and 2.5 to 4, respectively. Through the test of sounding experiment, the value of *T*_1_ is 2 and the value of *T*_2_ is 4 in this study.

The bathymetry residual values are obtained after fitting the trend surface with the polyhedral function. It is a fact that the proportion of contaminated bathymetric data collected would not be greater than 50% [34]. The matrix form $v^{\prime }$ of the standard residual $v_{i}^{\prime }$ for depth measurement expressed as:(11)$$\left\{\begin{array}{l} v^{\prime }=vv/\sigma_{\Delta v}\\ vv=\Delta v-med\{\Delta v\}\\ \sigma_{\Delta v}=med\{\left|vv\right|\}/0.6745 \end{array}\right.$$
where Δ*_V_* is the observed residual, obtained from Equation (8). *vv* is the normalized value of the sounding residual. $\sigma_{\Delta v}$ is the normalized median of non-zero absolute residuals.

Known nodes are regarded as independent and equal-precision observations. Each sounding point has a unit weight of 1 at the beginning of the iteration. Through the method of robust iteration, the fitting coefficient and the weight of observation value are continuously updated until $\underset{i}{\max}\left|C_{i}^{\left(k+1\right)}-C_{i}^{\left(k\right)}\right|<\varepsilon$ (*ε* is a small positive number) as the condition for stopping the iteration. The ultimate goal is to detect outliers with significant errors in the sounding dataset.

The IGGIII estimator’s data classification reveals that robust estimate is capable of producing reliable conclusions for locations with significant depth measurements. Additionally, the available data contains information with unusual sounds that has been given a lower weight to prevent it from affecting the model as a whole. Consequently, further detection is required for fine anomaly readings that are concealed in the existing data in order to accomplish high-precision quality estimation of seabed measurements.

### 2.3. Kriging Algorithm and RKA Theory

#### 2.3.1. Kriging Algorithm

It is assumed that a certain range of submarine topography is a second-order smooth change [28]. The estimation model of sounding data using kriging algorithm is:(12)$$\hat{L}\left(x_{0}\right)=\sum_{i=0}^{n}\lambda_{i}L_{i}$$
where $\hat{L}\left(x_{0}\right)$ is the estimated sounding value of undetermined point *x*_0_, *L_i_* is the sounding value of neighboring point *i*, and *λ_i_* is the contribution value of point *i* to the estimation.

Kriging algorithm is based on the variation function to obtain the estimated contribution value. The variation function between the two points is obtained by using the distance between the sounding points. The relationship matrix of weight *λ_i_* and Lagrange multiplier *μ* is constructed as:(13)$$\left[\begin{array}{cccc} \gamma_{11} & \cdots & \gamma_{n1} & 1\\ \vdots & \gamma_{ij} & \vdots & 1\\ \gamma_{n1} & \cdots & \gamma_{nn} & 1\\ 1 & \cdots & 1 & 0 \end{array}\right]\left[\begin{array}{c} \lambda_{1}\\ \vdots \\ \lambda_{n}\\ \mu \end{array}\right]=\left[\begin{array}{c} \gamma_{1b}\\ \vdots \\ \gamma_{nb}\\ 1 \end{array}\right]$$
where *γ_ij_* is the variation function value of the known node. *γ_ib_* is the variation function value of the estimated point *b* and the known node *i*.

Formula (13) is expressed in matrix form as:(14)$$\left[\begin{array}{cc} g & e\\ e^{T} & 0 \end{array}\right]\left[\begin{array}{c} \Lambda \\ \mu \end{array}\right]=\left[\begin{array}{c} g_{0}\\ 1 \end{array}\right]$$
where $e^{T}=\left[\begin{array}{ccc} 1 & \cdots & 1 \end{array}\right]$. $\Lambda ={\left[\begin{array}{ccc} \lambda_{1} & \cdots & \lambda_{n} \end{array}\right]}^{T}$. $g_{b}={\left[\begin{array}{ccc} \gamma_{1b} & \cdots & \gamma_{nb} \end{array}\right]}^{T}$.

The weight *λ_i_* and sounding estimation $\hat{L}\left(x_{0}\right)$ of the fixed point are obtained.
(15)$$\begin{array}{l} \lambda ={\left(e^{T}g^{-1}e\right)}^{-1}e^{T}g^{-1}L\\ \hat{L}\left(x_{0}\right)=\lambda +{g_{0}}^{T}g^{-1}\left(L-e\lambda \right) \end{array}$$

It can be seen that the quality estimation of depth measurement has been observed to be adversely affected in severe circumstances when aberrant points are included in the depth measurement dataset, hence contaminating the variation function fitting process of the Kriging algorithm. Consequently, the key to an improved Kriging algorithm is: (1) selection of variation function models; (2) resistance to variation function values.

#### 2.3.2. RKA Theory

Due to the irregular distribution of sounding data, we used more known sounding points to participate in the estimation of variation function. Based on the range of distance between points, (*m* + 1) equal spacing was set. $d_{l}=l\tau ,\;l=0,\;1,\;\ldots m$. *τ* is the segment length. Constant Δ*d* was selected, usually Δ*d = τ*/2. The water depth point pair with the distance condition $\left|d_{ij}-d_{l}\right|\leq \Delta d$ was searched and utilized. The variogram value for the sounding point distance *d_l_* is:(16)$$\gamma \left(d_{l}\right)=\frac{1}{2m_{c}}\sum {\left(L_{i}-L_{j}\right)}^{2}$$
where *m_c_* is the number of point pairs at the distance *d_l_*. *L_i_* and *L_j_* are the water depth values of sounding points.

However, the selection of known points inevitably contains a small number of abnormal sounding values in practice [36], which can be used to calculate the spatial structure of the variation function by Equation (16). Therefore, our improvement was to introduce polyhedral function and robust estimation to optimize the weights and centration values of these known points, without affecting the sample statistics and calculations of the mutation function. The derivation expression of robust kriging is:(17)$$\gamma \left(d_{l}\right)=\frac{1}{2}E\left({\left[L\left(x+d_{l}\right)-L\left(x\right)\right]}^{2}\right)=\frac{1}{2}E\left({\left[\left(L\left(x+d_{l}\right)-LL\right)-\left(L\left(x\right)-LL\right)\right]}^{2}\right)$$

The sample value of the variation function was obtained by using the centralized value of the sounding $\Delta L\left(x_{i}\right)=L\left(x_{i}\right)-\hat{L}{\left(x_{i}\right)}^{\left(k\right)}=L\left(x+d_{l}\right)-LL$. The variation function model of its robust estimation is:(18)$$\bar{\gamma }\left(d_{l}\right)=\frac{1}{2m_{c}}{\left[\sqrt{p_{i}}\Delta L\left(x_{i}\right)-\sqrt{p_{j}}\Delta L\left(x_{j}\right)\right]}^{2}$$

The variation function values of regular spacing are calculated based on the distribution and characteristics of known observed sample points. The relationship between *γ_ij_* and *d_l_* was analyzed to fit the model parameters. Subsequently, the actual values of variation function between known points, between estimated points and known points, were obtained. Theoretical fitting models of the variation function frequently employed the spherical and Gaussian models [28].

The spherical model expression is:(19)$$\gamma \left(d_{l}\right)=\left\{\begin{array}{cc} 0 & d_{l}=0\\ Ab_{0}+Ab\left(\frac{3d_{l}}{2r}-\frac{d_{l}^{3}}{2r^{3}}\right) & 0<d_{l}<r\\ Ab_{0}+Ab & d_{l}>r \end{array}\right.$$
where *Ab*_0_ is nugget value. *Ab*_(0)_ is denoted as abutment value. *Ab* represents offset abutment value. *r* is described variable range. *d_l_* is the regular spacing distance [29,37].

The Gaussian model expression is:(20)$$\gamma \left(d_{l}\right)=\left\{\begin{array}{cc} 0 & d_{l}=0\\ Ab_{0}+Ab\left(1-\exp(-\frac{d_{l}^{2}}{r^{2}})\right) & d_{l}>0 \end{array}\right.$$

Therefore, the fitting of the variation function was transformed into a multivariate nonlinear regression problem. According to the data pairs $\left[d_{l},\gamma_{d_{l}}\right]$ of sample variation function, the coefficients of nugget value *Ab*_0_, offset abutment value *Ab*, and variable range *r* were determined by least squares method. The variation function values corresponding to the spacing of each point could be obtained by substituting them into the determined model.

The estimated depth value at the designated point is calculated by Equation (15). Furthermore, the residual error is compared with the criterion of triple mean square error to achieve fine judgment of small outliers.

## 3. Experiments and Results

All code was based on MATLAB 2020a and run on the Windows 11 platform. Each module performance of DRS technique was analyzed through two sets and different terrains. The advantages of the proposed technique and module were reflected in fitting estimation, residual error, DEM effect map, contour map, and checkpoint indicator values.

### 3.1. Data Description and Parameter Settings

The multi-beam sounding dataset was derived from the Liwan 3-1 survey area in the South China Sea, where the depth of seawater is at least 1250 m. The Liwan 3-1 Gas Field Deepwater Drilling Platform is the largest deepwater drilling platform in China. Real time understanding of the seabed terrain around drilling is crucial for the stability of the platform. Two pieces of sounding data with different fluctuations were selected to analyze the detection performance of sounding outliers for estimating the quality of underwater terrain. Among them, the terrain of Area 1 is relatively more complex, with 20 Ping of sounding data. There is a total of 25 Ping depth measurement dataset in Area 2. The terrain is relatively flat with higher density, but there are relatively more anomalous values in depth measurements. Figure 2 shows the location of the measurement area and the information of collection equipment.

Some parameters for DRS technology were set as follows. The three elements of polyhedral function are: *δ* = 10,000, *β* = 0.5. *n* is half of the experimental observation data. According to the fluctuation of experimental terrain, the residual threshold of the IGGIII function is *T*_1_ = 2, *T*_2_ = 4 in the robust estimation model. The grid spacing of the variation function in RKA algorithm is set to 50 to ensure the intensiveness of data calculation.

Furthermore, we conducted a comparative analysis of four distinct schemes, namely RPF, WT, GF, and WLS-SVM, in order verify the efficacy and dependability of DRS’s anomaly detection technique. The important parameters and depth estimation criteria of the relevant schemes are described in Table 2.

### 3.2. The Value of Robust Estimation

This section analyzes sounding data from region 1, reflecting the importance of robust estimation. A total of 132 big sounding outliers are displayed in Figure 3 as the detection results of robust estimation for outliers. Scatter points, cluster spots, and inaccurate data with notable topographical variations can all be efficiently identified by DRS technology. Furthermore, as Figure 4 displays, the contour map exhibits a clearer effectiveness of eliminating outliers. The robust estimating procedure has resulted in reasonably smooth processed contour lines. Some sharp and protruding creases have been clearly removed. The significance of robust estimation is illustrated by these Figure 3 and Figure 4, which can precisely assess the quality of seabed topography and efficiently reduce the influence of sounding anomalies on changes in terrain.

### 3.3. Selection of Variation Function in RKA

The selection of variation function model is very important for the application of kriging algorithm [38,39]. In this section, Area 1 set was used for experiments. And 100 normal sounding points were selected for evaluation. Spherical model and Gaussian model were selected for comparative analysis, and the performance of RKA algorithm is shown in the optimal attitude, so as to achieve fine judgment of the sounding outliers.

Figure 5 illustrates that the Gaussian model can better fit the sample variation function values, which will help the kriging algorithm to effectively detect the sounding anomalies with small errors. The sum of nugget value (*Ab*_0_) and abutment value (*Ab*) falls as the distance between regular grids rises, which widens the gap between the theoretical fitting values of the variation function. The fitting effect and outlier identification are made more uncertain by this occurrence, which also makes it possible to tolerate minor error outlier locations that differ from our predicted outcomes.

Figure 6 displays the residual error of test points after applying different variation function fitting models. The residual error of gaussian model fluctuates within the permissible range. The spherical model’s residual inaccuracy is greater than 20 m in comparison to the sounding value of roughly 1300 m, and the variation is somewhat greater. It is advised against adopting the spherical model since the results show that it has a poor fitting effect on the variation function in the sounding data in this research. For this reason, the gaussian model was utilized in the ensuing tests to match the variation function.

### 3.4. Outlier Detection Effect of 1 Ping Sounding Data

The outlier detection capabilities of RPF model are demonstrated using three different approaches: wavelet analysis, robust polynomial fitting, and Gaussian filtering. 1 Ping sounding data with notable terrain undulations was chosen for the experiment. The 1 Ping data kept detection values with quite acceptable middle areas while removing scattering noise at both ends.

Peak anomalies of various sizes are clearly present in the original sounding data, as seen in Figure 7. We compared the quality of outlier detection among multiple schemes. Figure 7 reveals that robust polynomial fitting can only consider the trend of depth variation. Furthermore, robust polynomial fitting model prevents abnormal spots with characteristic changes from exhibiting excessive variance, particularly at the depth measurement locations on both ends. This model is overly rigorous and strict, which makes it easy to make mistakes in judgment and alter the significance of typical sounding locations. WT, GF, and RPF approach are the other three schemes that can all roughly match the waveform’s trend. By zooming in on the local area in Figure 7, it can be observed that our RPF model can effectively widen the gap with the peak outliers, which is more conducive to using weight functions or the criterion of triple mean square error for outlier detection.

The residual error is obtained by the difference between the sounding estimates and the observed values, as shown in Figure 8. The mean square error can characterize the removing performance of the model. While the mean square error of robust polynomial fitting is 0.176, the robust polynomial fitting mean square error is 0.0232. In addition, WT and GF are in the middle range. Additionally, the residual error fluctuation of WT is relatively large, which indirectly rises the value of mean square error and inevitably increases the uncertainty of determining the anomaly point. The residual fluctuation of GF is small at all positions, making it challenging to determine outliers using the criterion of triple mean square error. Besides, Figure 8 shows that robust polynomial fitting performs poorly in fitting depth measurements at both ends, resulting in excessive residuals in these areas and significantly leading to misjudgment of outliers. Meanwhile, the fitting values of our RPF model approximate the trend of sounding. The residual peak is directly noticeable in sounding anomalies, which helps track the anomaly’s source. Significant errors can be avoided when detecting anomalous soundings places with the RPF model.

### 3.5. Terrain Quality Estimation after Removing Outliers

The RPF, WT, GF, and WLS-SVM schemes were contrasted to demonstrate the excellent estimation of terrain quality by DRS technology. The Area 1 sounding dataset was employed to carry out this portion of the experiment. Additionally, the effect of terrain estimation can be illustrated by the DEM and contour map after removing outliers.

Figure 9 shows that five strategies for eliminating outliers can effectively suppress the interference of sounding outliers on terrain. The terrain processed by RPF, WLS-SVM, and DRS technologies is relatively smooth. There are jagged concerns with the terrain that WT and GF processed. Besides, WT and GF are limited in their ability to handle certain peak signals for cluster anomalies at gullies. Simultaneously, the WLS-SVM fitting judgment on the gully and terrain back needs to be strengthened. The DRS approach successfully minimizes interference from identifying outliers while also permitting fluctuations in typical sounding spots. It can precisely assess the quality of the terrain and display and restore regions with deep shadows and other changes in the ground. Moreover, the advantages of suggested DRS technology are evident in the local feature terrain anomalies, multi variation outliers of cluster type, and huge error outliers.

Figure 10 shows the contour map results after removing outliers from each strategy. A significant amount of smoothing has been done to the bathymetry, however cluster-shaped outliers still exist in the middle and both ends of the RPF edge processing. Large outliers can be found with WT and GF, but tiny errors are dispersed throughout the detection region. The contour lines that have been treated with WT and GF also have a sawtooth shape, which suggests that there are still unresolved outliers present. There are tiny mistakes (circles) in the WLS-SVM results. DRS technology is a dual detection and judgment method for outliers in depth measurement datasets. With regard to discrete distribution anomalies, cluster anomalies, and intricate terrain changes, DRS technology can accurately identify sounding spots with significant errors and tiny anomalies.

## 4. Discussion

### 4.1. The Impact of Different Terrains and Distribution of Anomaly Points

In order to verify the response of the proposed DRS technology to different terrains and anomaly distributions, we used the RPF, WT, GF, and WLS-SVM schemes to assess the terrain quality. The Area 2 sounding dataset with higher terrain complexity and wider error distribution was analyzed. Furthermore, a sample of 100 typical sounding points was chosen for a discussion on the suggested technology’s capacity to solve problems. The performance of errors and residuals, as displayed in Table 3 and Figure 11, shows the excellent performance of the proposed DRS technique.

Table 3 indicates that DRS technology’s exterior accuracy on the test point set is 0.8321. Furthermore, the test points’ fluctuation range is likewise the smallest. The WT and GF fitting errors are non-symmetrical and have a relatively high accuracy. The residuals of 100 test points chosen at random are displayed in Figure 11. The residual variation is highest for WT and GF, with a wide distribution of positive values. The performance of WLS-SVM and RPF comes next. With residuals managed within 2 m, the DRS technology provides a satisfactory fitting effect at an average depth of 1300 m. The excellent results of DRS technique, which can approach the seabed for topography mapping and indirectly detect and find anomalous locations in depth measurement, is completely reflected in Table 3 and Figure 11.

The original DEM map has numerous sharp sounding abnormalities, as shown in Figure 12a. There are a lot of unusual sounding spots in this graph that are concentrated and uniformly dispersed. It was discovered by subjective visual observation that the landscape treated with WT and GF had jagged features, which amply demonstrated the existence of minor errors with discontinuous distributions that were not eliminated. Numerous clustering outliers, which are distinguished by deep troughs and local protrusions, are also included after outliers in RPF and WLS-SVM have been eliminated. Large error discrete detection spots can be successfully removed by the suggested DRS technology, while tiny error detection outliers in the cluster can be batch processed.

In addition, Figure 13 displays that the contour map results processed by RPF, WT, GF, and WLS-SVM exhibit certain defects (circles) to varying degrees. The continuity of contour lines, residual sounding anomalies, and extension of sounding can fully demonstrate that the proposed DRS technology can ensure effective detection and correct removal of sounding anomalies.

### 4.2. Performance Analysis of Waveform Filtering and Trend Surface Construction Models

The recommended waveform filtering techniques operate under the presumption that the seabed landscape varies gradually and continuously. As can be seen from the WT and GF treatment effects in Figure 7, Figure 9 and Figure 10, the anti-fluctuation capacity may not be outstanding. With the exception of polynomial fitting techniques, the trend surface construction approach can be used in regions with very complex topography. There is an issue with inadequate outlier removal at the edges for waveform filtering operations based on coordinate grids and sliding along or perpendicular to the trajectory direction because of a lack of data. Furthermore, regions with notable topography fluctuations, cluster outliers, and minor error outliers are unable to be well resolved by WT and GF waveform filtering techniques. The trend surface building model will ignore the outlier points with modest errors if it fails to employ weighted processing or robust estimation and instead evaluates outliers using the three times mean square error criterion. Even with the addition of reliable estimating modules, there is still a lack of data, and careful judgment and detection are needed. As a result, the RKA module is expanded upon in the paper to enhance the identification of minute error outliers. The performance advantage is effectively highlighted in the final renderings of Figure 9, Figure 10, Figure 12 and Figure 13.

## 5. Conclusions

This paper presents a dual robust strategy (DRS) technique to achieve high-precision assessment of seabed terrain quality by combining robust polyhedral function (RPF) and robust Kriging algorithm (RKA) to remove outliers in depth measurements. The fitting effect of the 1 Ping sounding, the value of robust estimation, and the selection fitting model of variation function were all examined independently in the study. Robust estimation experiments have shown that robust estimation can quickly reduce the interference of sounding anomalies on seabed topography. Furthermore, the Area 1 experiment demonstrates that the Gaussian model performs better when calculating variational functions, which accurately illustrates the RKA algorithm’s modest error detection capabilities. The fitting value of RPF is more in line with the actual seabed terrain and is more effective in identifying greater aberrant depth readings, according to the results of 1 Ping sounding data. Finally, techniques like WT, GF, and WLS-SVM were applied to prove the superiority and dependability of the suggested DRS technology. With DRS technology, outlier interference on depth estimation may be minimized and the true seabed terrain can be approached steadily. It can detect anomalous information at discrete places, cluster points, and areas where the undulation of the landscape changes because to its dual detection capacity, which guarantees its flexibility to various terrain difficulties.

## Figures and Tables

**Figure 1 sensors-24-01476-f001:**
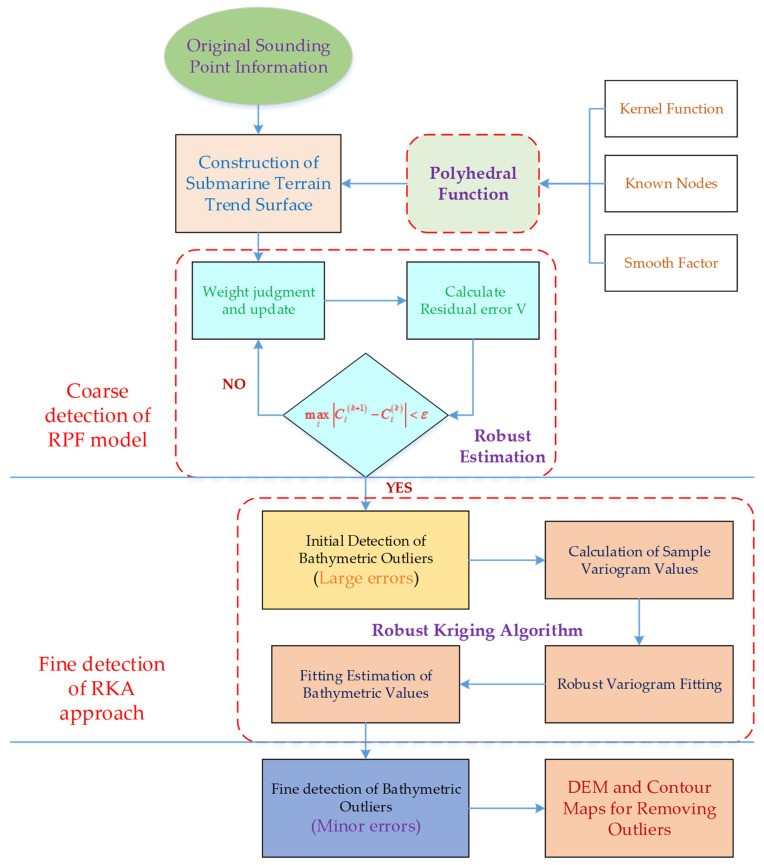
Flow chart of DRS technique for filtering sounding outliers to improve seabed terrain quality estimation.

**Figure 2 sensors-24-01476-f002:**
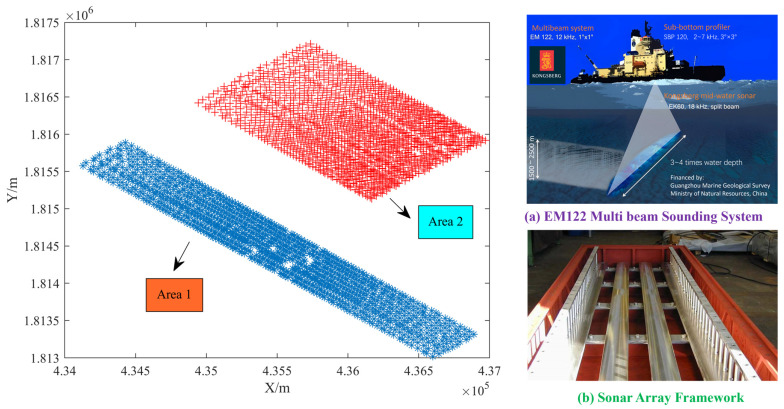
Experimental data location and collection equipment information.

**Figure 3 sensors-24-01476-f003:**
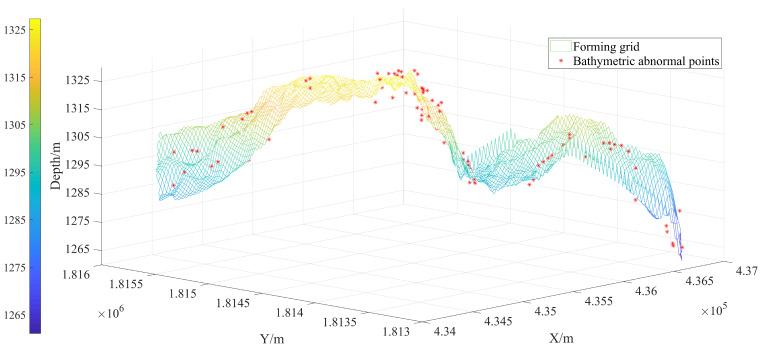
Outlier detection effects of robust estimation in the DRS technology.

**Figure 4 sensors-24-01476-f004:**
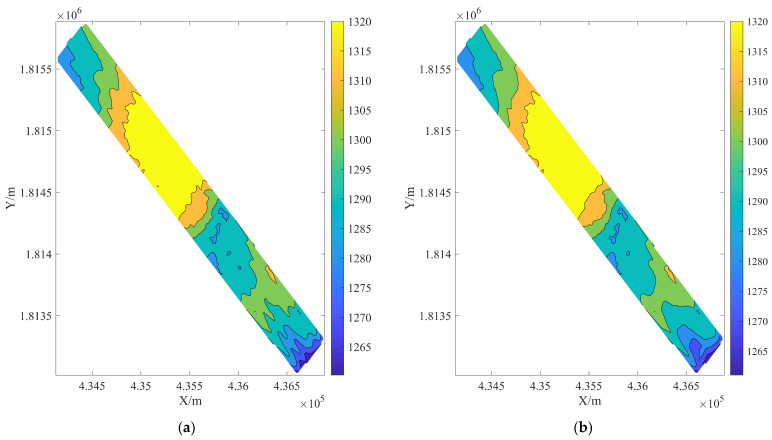
The effect of contour map after removing outliers. (**a**) original bathymetry set. (**b**) the results of robust estimation in the DRS technology.

**Figure 5 sensors-24-01476-f005:**
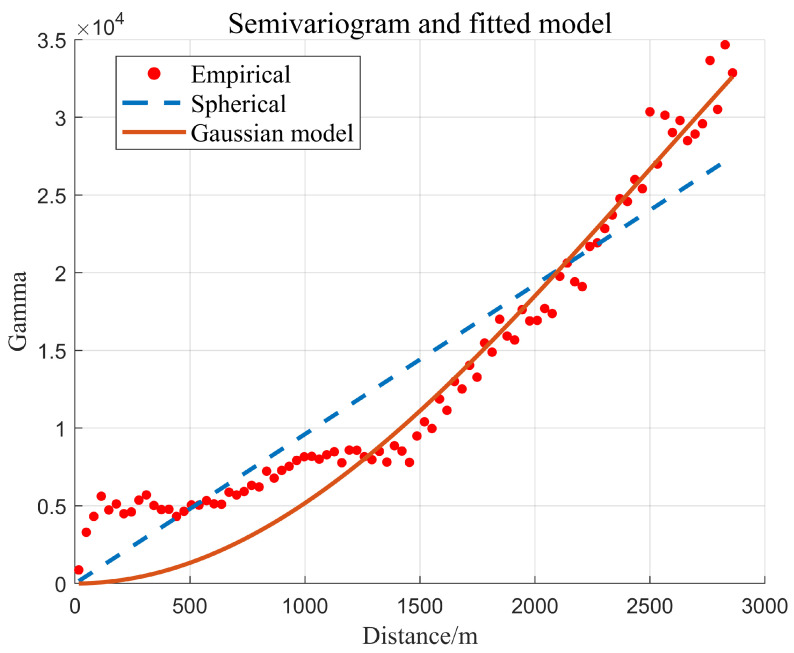
The fitting effect of different variation models in the RKA approach.

**Figure 6 sensors-24-01476-f006:**
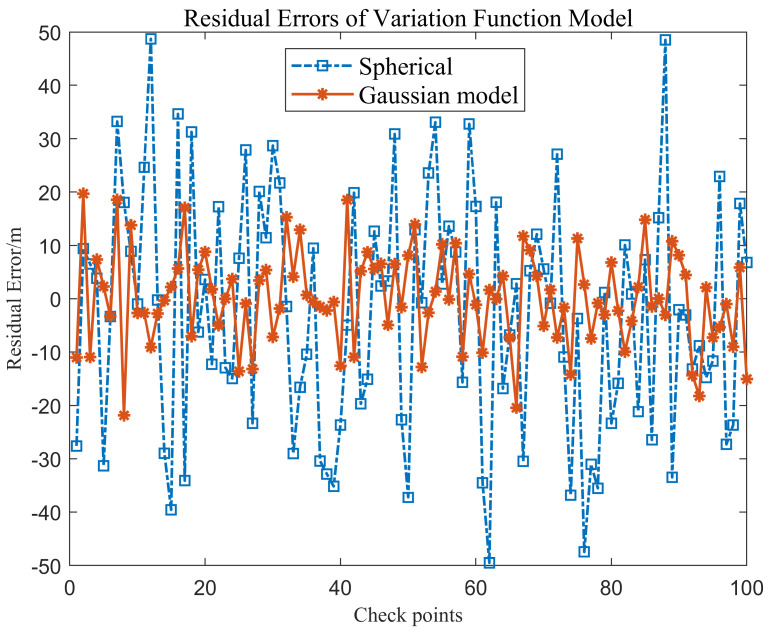
The residual error of the variogram model for the validation set.

**Figure 7 sensors-24-01476-f007:**
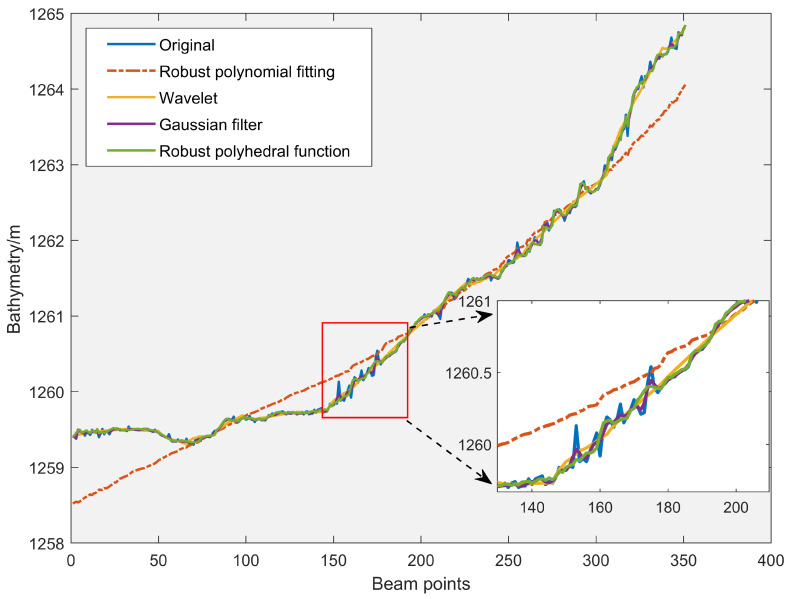
The fitting effect of 1 Ping bathymetry data.

**Figure 8 sensors-24-01476-f008:**
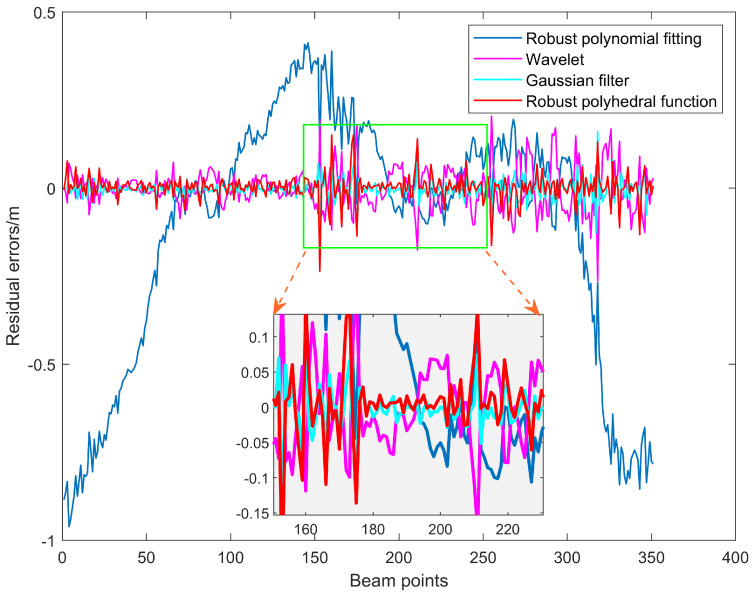
Fitted residual errors of 1 Ping bathymetry data.

**Figure 9 sensors-24-01476-f009:**
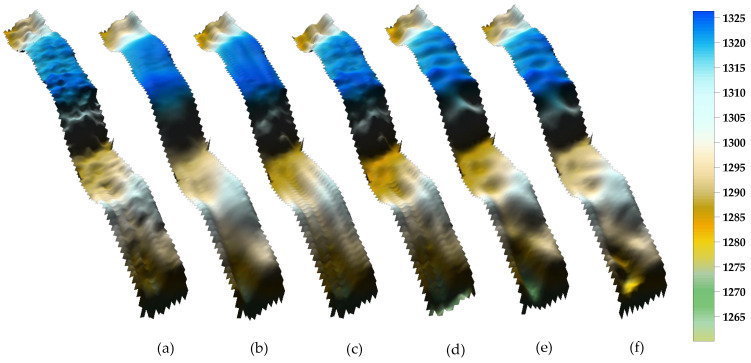
DEM maps of Area 1 after eliminating outliers. (**a**) original DEM. (**b**) RPF. (**c**) WT. (**d**) GF. (**e**) WLS-SVM. (**f**) DRS technology.

**Figure 10 sensors-24-01476-f010:**
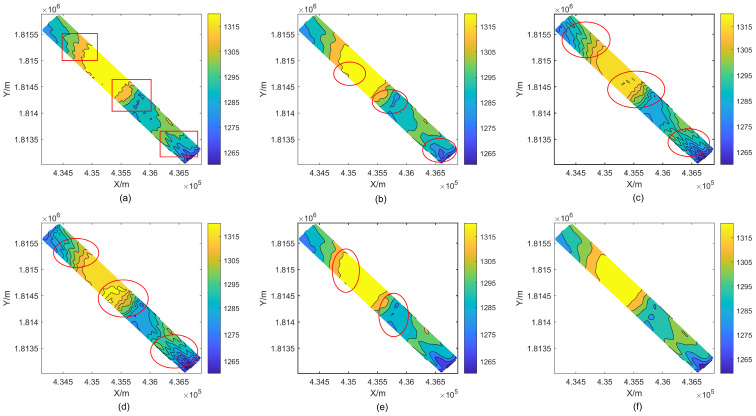
Contour maps of Area 1 after removing outliers. (**a**) original data. (**b**) RPF. (**c**) WT. (**d**) GF. (**e**) WLS-SVM. (**f**) DRS technology. The red boxes and circles in the figure are used to highlight the experimental effect in this area.

**Figure 11 sensors-24-01476-f011:**
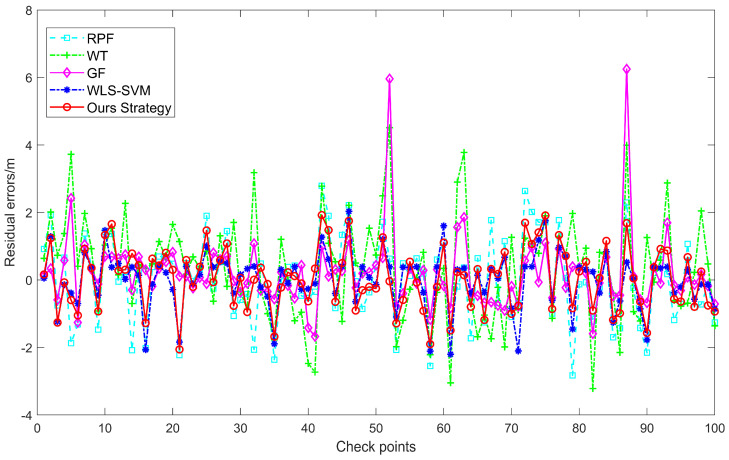
Residual errors of test set in Area 2.

**Figure 12 sensors-24-01476-f012:**
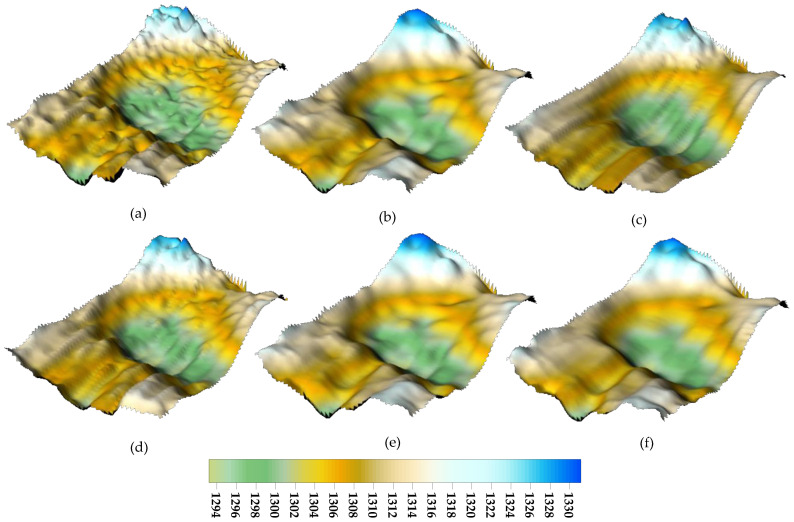
DEM maps of Area 2 after removal of bathymetric outliers. (**a**) original DEM. (**b**) RPF. (**c**) WT. (**d**) GF. (**e**) WLS-SVM. (**f**) DRS technology.

**Figure 13 sensors-24-01476-f013:**
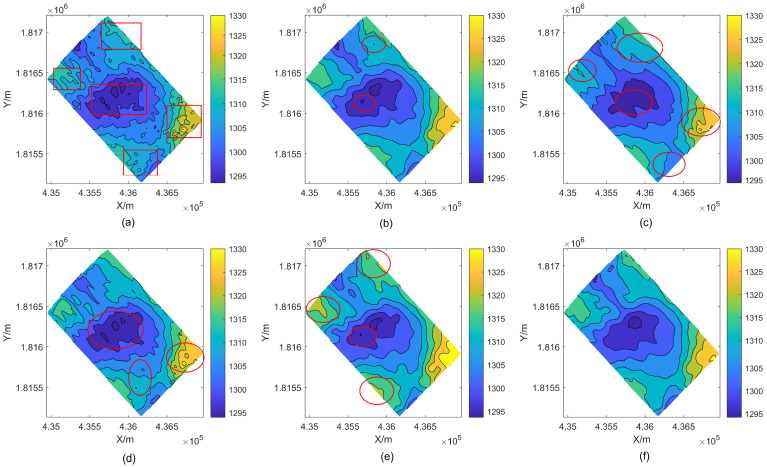
Contour maps of Area 2 after removal of bathymetric outliers. (**a**) original DEM. (**b**) RPF. (**c**) WT. (**d**) GF. (**e**) WLS-SVM. (**f**) DRS technology. The red boxes and circles in the figure are used to highlight the experimental effect in this area.

**Table 2 sensors-24-01476-t002:** Explanation of important parameters and depth measurement criteria for each scheme.

Schemes	Parameter Description	Outlier Evaluation Criteria	
		Weight Function	Triple Mean Square Error
RPF	*T*_1_ = 2; *T*_2_ = 4	√ ^1^	
WT	The wavelet basis function is ‘db2′, with a decomposition level of 4.		√
GF	Gaussian template size is 5; The noise standard deviation is the unit weight median error of bathymetric data.		√
WLS-SVM	Coordinate *x* and *y* as feature inputs; The kernel function type t is a radial basis function; The penalty coefficient *C* =1; The loss function parameter *p* = 0.01.		√
Our DRS technique	*δ* = 10,000, *β* = 0.5; The number of known nodes *n* is half of the depth measurement dataset; *T*_1_ = 2; *T*_2_ = 4; The width between grids was set to 50.	√	√

^1^ The symbol √ indicates that the evaluation criterion has been added to the technical scheme.

**Table 3 sensors-24-01476-t003:** Comparison of external accuracy (unit: m).

	RPF	WT	GF	WLS-SVM	DRS
External accuracy	1.2615	1.5502	1.1022	0.9229	0.8321
Minimum error	−2.826	−3.2166	−1.6758	−2.2022	−2.0582
Maximum error	2.7886	4.5137	6.2481	2.0312	1.9209

## Data Availability

The experiment uses an internal dataset and the data presented in this study are available on request from the corresponding author.

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
