# Peer review of "A Dual Robust Strategy for Removing Outliers in Multi-Beam Sounding to Improve Seabed Terrain Quality Estimation"

_sensors, 2024, doi:10.3390/s24051476_

Round 1
Reviewer 1 Report
Comments and Suggestions for Authors
Comments:
This paper focuses on removing outliers in multibeam sounding to improve its underwater terrain quality. The authors use a polyhedral function to fit the trend surface, use robust estimation to weaken the effects of abnormal values and use robust kriging algorithm to detect small outliers. This paper also evaluated the proposed technology using its own dataset. The paper has clear logic and detailed content, but there are still some details that can be improved.
If the author can deal with the following questions, it may be considered for publication.
1. This paper requires a reorganization of the abstract, adding a better discussion of existing related work and the motivation for improvement, emphasizing the contribution of their technology in this field compared to other methods.
2. The method used in this paper includes three steps: determining the trend surface, adjusting the robust estimation of weights, and DEM interpolation construction. Therefore, when introducing relevant research, only the algorithms that can be selected when executing these three steps were introduced. Other related research still needs to be introduced, including the gaps between existing research and proposed solutions.
3. Due to that the dual robust estimation method is not proposed by the author, it is recommended to strengthen the advantages or characteristics of the dual method in this paper, rather than simply using “novel” to explain.
4. The academic contribution of this paper has not been clearly demonstrated, and better writing is needed in the algorithm explanation and conclusion. The current writing style may make readers believe that the author has chosen three models or functions from each of the three steps of the algorithm framework, combined into one algorithm, which will have a negative impression on the actual contribution of this paper.
5. The writing of the article should be more rigorous. The titles on lines 142 and 151 are repetitive, and the content on lines 318-329 and 304 to 315 is also completely repetitive.
Comments on the Quality of English LanguageThe writing of the article should be more rigorous. Repetitive content is considered careless and may leave an impression of not being careful enough.
Author Response
- This paper requires a reorganization of the abstract, adding a better discussion of existing related work and the motivation for improvement, emphasizing the contribution of their technology in this field compared to other methods.
Response: It is a good suggestion. The abstract of this article has been rewritten. Firstly, it is inevitable to obtain anomalous sounding values during the process of multi beam bathymetry. Then, the existing problems in the current research were expressed. And the proposed technology was discussed. Finally, the experimental conclusion and significance were elaborated in detail.
- The method used in this paper includes three steps: determining the trend surface, adjusting the robust estimation of weights, and DEM interpolation construction. Therefore, when introducing relevant research, only the algorithms that can be selected when executing these three steps were introduced. Other related research still needs to be introduced, including the gaps between existing research and proposed solutions.
Response: Thank you for your comments. The introduction of this article has been rewritten. The detection of sounding anomalies and terrain quality evaluation are mainly discussed from the perspectives of manual editing, waveform filtering, surface fitting, and robust estimation, as shown in Table 1.
Table 1. Statistical Methods for Eliminating Abnormalities in Sounding to Improve the Quality Estimation of Submarine Topography.
|
Aspects |
Techniques |
Characterization |
|
Manual editing |
Main software: Qimera; Hypack; PDS; CARIS; MB-System; etc. [30] |
Time-consuming and rough elimination. |
|
Waveform filtering |
WMA [9]; GF [10]; WT [11]; Kalman filtering [12]; etc. |
In units of Ping signal; Relatively low efficiency; Outliers of significant features. |
|
Surface fitting |
Polynomial fitting [13]; CUBE [14]; WLS-SVM [15]; B-splines [16]; Polyhedral function [17-19]; etc. |
Simple structure; Outliers of significant features; Susceptible to outlier contamination. |
|
Robust estimation |
Huber function [21]; L1 norm [22]; IGGIII estimator [23, 25]; Tukey test [24]; etc. |
Pollution-resistant; High accuracy; Outliers of significant features. |
|
Interpolation construction |
inverse distance weighting [26]; nearest neighbor interpolation [27]; kriging algorithm [28-29]; etc. |
Simple structure; Requires high precision processing up front. |
- Due to that the dual robust estimation method is not proposed by the author, it is recommended to strengthen the advantages or characteristics of the dual method in this paper, rather than simply using “novel” to explain.
Response: It is a good suggestion. This article focuses on the sources, ideas, advantages, and improvement processes of robust estimation, as detailed in sections 2.2 and 2.3.
4 The academic contribution of this paper has not been clearly demonstrated, and better writing is needed in the algorithm explanation and conclusion. The current writing style may make readers believe that the author has chosen three models or functions from each of the three steps of the algorithm framework, combined into one algorithm, which will have a negative impression on the actual contribution of this paper.
Response: Thank you for your comments. The writing style of this article has been revised. The main focus of the experiment and analysis is to explore the quality of anomaly detection, robustness estimation, and optimal selection of the mutation function for 1 ping. The discussion section mainly introduces the performance comparison between waveform filtering and trend surface, as well as the impact of outlier distribution and terrain complexity on the algorithm.
- The writing of the article should be more rigorous. The titles on lines 142 and 151 are repetitive, and the content on lines 318-329 and 304 to 315 is also completely repetitive.
Response: It's our fault to ignore the problem. The relevant content has been deleted and rewritten.

Reviewer 2 Report
Comments and Suggestions for Authors
See the attached file.

English is good. A few improvements are needed.
Author Response
Thank you very much for providing valuable feedback. Please refer to the document for relevant modification suggestions.

Reviewer 3 Report
Comments and Suggestions for Authors
Author Response

(The authors gave the same response as above.)
